# Impact of genomic literacy components on genetic testing decision-making in the general Japanese population in the 20s and 30s

Tomomi Miyoshi[1]☯, Masaki Watanabe[2]☯*

**1** Department of Cardiovascular Medicine, The University of Tokyo Hospital, Tokyo, Japan, **2** Graduate School of Teacher Education, Tokyo Gakugei University, Tokyo, Japan

☯ These authors contributed equally to this work.
* masawata@u-gakugei.ac.jp

**Data Availability Statement:** The data that support the findings of this study are openly available at: https://figshare.com/, reference number (10.6084/m9.figshare.22122380).

## Abstract

In promoting genomic medicine, genomic literacy, which is the public's ability to acquire and utilize the knowledge and skills related to genetics and genomes, requires urgent improvement. This study determined how the three components of genomic literacy (genetic/genomic knowledge, health numeracy, and interactive and critical health literacy) are associated with risk and benefit perception for genetic testing and decision-making related to genetic testing. Using an online questionnaire, we surveyed genetic/genomic knowledge, health numeracy, interactive and critical health literacy, risk and benefit perception of genetic testing, and intention toward genetic testing of 857 Japanese citizens (aged 20–39). A vignette was created to measure the intention toward genetic testing, and respondents were asked about their willingness to undergo genetic testing and to share the results with their partners and children. The path analysis, which was done by creating a path diagram revealing the relationship of the three components with risk and benefit perception, revealed that genetic and genomic knowledge and interactive and critical health literacy affected the respondents' intention to undergo genetic testing via recognition of the benefits of such testing. Further, it was suggested that health numeracy affects benefit perception through risk perception toward genetic testing. However, the goodness of fit of this model for the intention of conveying positive results to partners and children was relatively low, thus indicating that the hypothetical model needs to be reexamined.

## Introduction

In recent years, expectations for genomic medicine have rapidly increased with advancements in genetics and genomics. For example, in the field of cancer and intractable diseases, the practical application of genomic medicine, such as more effective and efficient diagnosis and treatment of diseases suitable for an individual's constitution and medical condition, has begun [1]. Meanwhile, with the rapid market expansion of Direct-to-Consumer Genetic Testing business and the associated media coverage, the general public has more opportunities to encounter words such as "genomes" and "genes." Currently, a person can attain their own genome information, which requires the ability to have and utilize knowledge about genetics and genomes.

**Funding:** The authors received no specific funding for this work.

**Competing interests:** The authors have declared that no competing interests exist.

However, it is difficult for the general public to fully understand genomic medicine due to its complexity and the high level of technological expertise required; as such, the general public's knowledge of heredity and genomics is insufficient [2, 3]. Furthermore, the genetic and genomic information provided through the media is not always correct, and many people find it difficult to distinguish between valid and invalid genetic and genomic information [2, 4]. Therefore, there exist excessive expectations and misunderstandings, and also concerns that the acquisition of biased genetic and genomic information and incorrect knowledge may lead to inappropriate behavior.

It has been noted that knowledge about genetics and genomics alone does not lead to appropriate action. Based on the concept of health literacy, the American Public Health Association [5] states that the general public needs to acquire the knowledge and skills necessary to incorporate genetic and genomic information into health-related decision-making. The National Human Genome Research Institute [6] also states the need for genomic health literacy to enhance individual decision-making capacity. In the field of genetic and genomic research, people with low health literacy may have problems understanding and using genetic and genomic information and, consequently, may choose risky health behaviors. Definitions of genomic literacy based on the concept of health literacy include numeracy and interactive and critical health literacy in addition to genetic/genomic knowledge [7, 8]. According to Nutbeam [9], interactive health literacy is the skill of using new information in response to changing situations, while critical health literacy is the skill of critically analyzing information. Furthermore, numeracy plays an important role in interpreting the risks and benefits of genomic medicine and making appropriate decisions. Therefore, in addition to genetic/genomic knowledge, numeracy, interactive and critical health literacy are important components of genomic literacy [10].

Typical health behavioral theories, such as the Health Belief Model and Protection Motivation Theory, demonstrate that the perception of risks and benefits is an important determinant of decision-making [11]. Health education intended for behavioral change often incorporates disease risk awareness and coping benefit awareness into educational goals [11]. Many studies support the impact of risk and benefit perception on health decision-making [12–15]. In the area of genetic counseling, risk perception is also positioned as an important element [16]. Therefore, risk and benefit perception of test and treatment methods are considered to be important determinants for decision-making in the genetic and genomic domain.

According to a study of the processes by which risk and benefit perceptions influence decision-making, the following association has been found in the risk perception process: general tendency → situation recognition factor → option recognition factor → behavior. This suggests that the decision-making process in which each individual's abstract way of thinking and recognizing influences the recognition of a concrete object, and as a result, determines their behavior, exists securely [17]. Furthermore, it has been confirmed that knowledge and critical thinking define intention through risk and benefit perception as determinants of acceptance of risk [18]. It has also been pointed out that numeracy, which is an important element of health behavior, affects risk and benefit perception [19]. Therefore, in this study, we assumed a process in which the three components of genomic literacy affect decision-making through risk and benefit perception and examined their effects on decision-making.

Although genomic medicine is becoming widespread in Japan, there are still few opportunities for the general public to make decisions, because related genetic tests and treatments are not yet commonly known. In Japan, decision-making research on genetic testing is limited to the perception of patients undergoing genetic testing and their decision support [20, 21]. Therefore, this study presented a virtual scene in the form of a vignette to evaluate decision-making in situations related to taking genetic tests. Prior studies have indicated that young

people have inadequate knowledge of general genetics and genomics and do not accurately estimate the risks involved [22]. Therefore, we surveyed members of the general public in their 20s and 30s, who were likely to be exposed to genomic medicine in the near future.

Based on the above understanding, this study considers genetic/genomic knowledge, health numeracy, and interactive and critical health literacy as components of genomic literacy. It examines the impact of risk and benefit perception on decision-making relating to genetic testing.

## Materials and methods

### Survey respondents

The survey covered 2,000 respondents (1016 males and 984 females) between the ages of 20 and 59, and 857 respondents (438 males and 419 females) between the ages of 20 and 39 were included in the analysis of this survey. Since it has been noted that online surveys can bias respondents based on factors such as gender and age, the number of respondents for this survey was determined after evenly stratifying the gender and age groups according to the population distribution and percentages in the 2019 national survey. This study was approved by the Research Ethics Review Board of the Tokyo Gakugei University with which the authors are affiliated (approval number: 449).

The purpose of the study and the handling of personal information were explained on the webpage, and responses were considered to be the acquisition of consent.

### Questionnaire items

**Attributes.** The respondents were asked about their sex, age, marital status, children, occupation, level of education, whether or not they had taken biology during high school, whether or not they had learned about genetics, and whether or not they or someone in their family had a genetic disorder.

**Genetic/Genomic knowledge.** It has been pointed out that the existing genetic and genomic knowledge scales are biased toward assessing professional and academic knowledge [23]. Therefore, in this study based on the surveys and scales of previous studies [24–27], 11 items of genetic and genomic knowledge necessary for decision-making on genetic testing were created based on the advice of seven genomic medicine doctors and researchers. These eleven items were "characteristics of genetic information" (three questions), "relationship between gene mutation and disease" (three questions), and "knowledge of genetic testing" (five questions). In each question, the respondent was asked to select one answer from three options: "yes," "no," and "unsure." "Yes" was assigned one point, "no" or "unsure" was assigned 0 points, and the total score of genetic/genomic knowledge was considered as the "genetic/genomic knowledge score" (score range: 0 to 11 points).

**Health numeracy.** Health numeracy was measured using the validated scale "the health numeracy test" by Miyoshi et al. [28]. The questions in the health numeracy test are categorized as follows: "% and magnitude of risk" (four questions), "conversion from frequency to %" (one question), "calculation of probability" (one question), "conversion from probability to frequency" (two questions), and "reading tables and graphs" (two questions). "Yes" was assigned 1 point, "no" or "unsure" was assigned 0 points, and the total score of the 10 items in the health numeracy test was considered as the "health numeracy score" (score range: 0 to 10 points).

**Interactive and critical health literacy.** Japan is a country with high literacy, and the Communicative and Critical Health Literacy (CCHL) scale [29], developed by Ishikawa et al., is often used to measure interactive and critical health literacy (a higher level of health literacy

than functional health literacy). In this study, the measurement was carried out using the CCHL scale, which consists of five items. Responses are based on a five-point scale with "strongly agree" (5 points) as the highest and "strongly disagree" (1 point) as the lowest. The total score of the CCHL scale is considered as the "CCHL score" (score range: 5 to 25 points).

**Risk perception and benefit perception for genetic testing.** Based on the "Guidelines for Genetic Testing/Diagnosis in Medicine" [30], we created items for risk and benefit perception of genetic testing.

The content of questions on risk perception for genetic testing consists of five items such as "if I discover a genetic mutation (genetic change) associated with a disease, I will continue to be anxious about developing it in the future," and "if I discover a genetic mutation (genetic change) associated with a disease, it will put a strain on my family and future children."

The content of the questions on benefit perception for genetic testing consists of five items such as "if I discover a disease-related gene mutation (genetic change), it will help me prevent, detect early, and choose treatment for my own disease," and "if I discover a genetic mutation (genetic change) related to a disease, it will help prevent, diagnose early, and treat the disease in my family and future children." The response to each question is based on a five-point scale with "strongly agree" (5 points) as the highest and "strongly disagree" (1 point) as the lowest value.

**Genetic testing decision-making.** Although genomic medicine is becoming widespread in Japan, there are still few situations in which the general public makes decisions of their own, because genetic tests and treatments are not commonplace. Therefore, in measuring behavioral intention toward genetic testing, we used a method of assessing behavioral intention by presenting a situation regarding genetic testing in the form of a vignette; a vignette assumes family medical history and genetic risk.

The vignettes were created based on the advice of seven genomic medicine doctors and researchers. In this study, behavioral intention was regarded as decision-making in a narrow sense, and the term "behavioral intention" was used for measuring decision-making.

**Vignette for respondents.** "You are in your twenties and engaged. Your mother got sick at the age of 40. Your grandmother had also developed the same illness. You are in good health now. It is said that in 5–10% of cases, this disease is hereditary. If a genetic test is performed and a positive diagnosis is made, the probability of developing the disease in your lifetime is said to be 25–85%."

Regarding the question items on behavioral intentions, the question on "the respondent's genetic testing intention" is as follows: "Do you plan on taking a genetic test for this disease?" The response to this question is based on a five-point scale with "I will take the test" (5 points) as the highest and "I will not take the test" (1 point) as the lowest. The question on "intention to convey positive results to my partner" is as follows: "The genetic test revealed that you were positive for a specific genetic mutation. Would you share this result with your partner?" The question on "intention to convey positive results to your children" is as follows: "The genetic test revealed that you were positive for a specific genetic mutation. Would you tell this result to your future children?" The responses to both questions are based on a five-point scale with "I will tell them" (5 points) as the highest and "I will not tell them" (1 point) as the lowest.

## Analysis method

In the comparison of the average scores of the three components of genomic literacy, risk perception, benefit perception, and behavioral intention toward genetic testing by attribute, the effect size d was obtained by conducting a t-test on sex, age group, marital status, children, occupation, whether the respondent took biology in high school, whether the respondent had

an educational background in genetics, and whether the respondent or a family member had a genetic disease. For any level of education, a one-way ANOVA was performed to obtain the effect size $\eta^2$. Next, a correlation analysis was conducted to examine the relationship between the three genomic literacy variables, risk perception score, benefit perception score, and behavioral intentions toward genetic testing. To investigate the effects of genomic literacy components on behavioral intention toward genetic testing, those who had a genetic disorder or a family member with a genetic disorder were excluded and a multiple regression analysis was performed, with behavioral intention towards genetic testing as the dependent variable and the three components of genomic literacy as the independent variables. Next, we created a path diagram demonstrating the relationship between the three components of genomic literacy centered on behavioral intention toward genetic testing and the risk and benefit perception of genetic testing and performed path analysis. The values of GFI, AGFI, CFI, and RMSEA were calculated as indicators of the goodness of fit of the model. The significance probability of all analyses was set to the 5% level (both sides), and IBM SPSS Statistics 25 and IBM SPSS Amos 22 were used for the statistical analyses.

## Ethical considerations

The research company that conducted this survey has obtained the Privacy Mark and ISO/IEC27001 (ISMS) certifications from the Japan Information Processing Development Corporation (JIPDEC), and is an organization that gives due consideration to the protection of registered users' personal information, and has obtained the Privacy Mark and ISO/IEC27001 (ISMS) certifications. The personal information of the monitors is managed by the research company, and no more personal information than necessary will be obtained by the researcher. This study has been approved by the University's Research Ethics Committee (approval number: 449). The purpose of the study and the handling of personal information were explained on the webpage, and responses were considered to be the acquisition of consent.

## Results

### Attributes

At the time of the survey, the responses were collected after excluding those who responded to the straight line; thus, a total of 857 respondents were included in the analysis. The attributes of the subjects are shown in Table 1. The mean age of the respondents was 30.5 years (standard deviation: 5.40).

### Comparison of components of genomic literacy by attribute

The percentages of correct answers for genetic and genomic knowledge are shown in Table 2. Table 1 provides a comparison of the average scores of the three components of genomic literacy by attribute. The average genetic/genomic knowledge score was 5.67 points (standard deviation: 3.57; range: 0–11 points). As a result of comparing genetic/genomic knowledge scores by attribute, those with a high level of education, medical professionals, those who took biology in high school, those who had studied genetics, and those who had a genetic disorder or a family member with a genetic disorder had significantly higher genetic/genomic knowledge scores (p < .05). The average score for health numeracy was 7.11 points (standard deviation: 2.78; range: 0–10 points). As a result of comparing the health numeracy scores by attribute, the health numeracy scores were significantly higher in men and those with a high level of education. The average CCHL score was 17.84 points (standard deviation: 3.68; range: 5–25 points).

**Table 1. Comparison of components of genomic literacy by attribute.**

| | | N | % | genetic/genomic knowledge score | | t | p (effect size) | Health numeracy score | | t | p (effect size) | CCHL score | | t | p (effect size) |
|---|---|---|---|---|---|---|---|---|---|---|---|---|---|---|---|
| | | | | M | SD | | | M | SD | | | M | SD | | |
| Sex[1] | male | 438 | 51.1 | 5.75 | 3.57 | 0.69 | .489 (0.05) | 7.31 | 2.74 | 2.12 | .034 (0.14) | 18.02 | 3.75 | 1.36 | .173 (0.09) |
| | female | 419 | 48.9 | 5.58 | 3.57 | | | 6.91 | 2.80 | | | 17.67 | 3.60 | | |
| Age group[1] | 20s | 395 | 46.1 | 5.70 | 3.58 | 0.23 | .820 (0.02) | 7.09 | 2.81 | -0.28 | .078 (0.02) | 17.99 | 3.75 | 1.02 | .307 (0.07) |
| | 30s | 462 | 53.9 | 5.64 | 3.56 | | | 7.14 | 2.75 | | | 17.73 | 3.62 | | |
| Marital status[1] | Married | 424 | 49.5 | 5.55 | 3.49 | 1.18 | .239 (0.08) | 7.07 | 2.72 | 0.74 | .460 (0.05) | 17.81 | 3.49 | 0.53 | .596 (0.04) |
| | Unmarried, divorced or widowed | 428 | 49.9 | 5.84 | 3.62 | | | 7.21 | 2.80 | | | 17.94 | 3.80 | | |
| Children[1] | Yes | 321 | 37.5 | 5.57 | 3.47 | 0.63 | .528 (0.04) | 6.97 | 2.79 | 1.19 | .235 (0.05) | 17.90 | 3.69 | -0.34 | .735 (0.04) |
| | No | 536 | 62.5 | 5.73 | 3.63 | | | 7.20 | 2.77 | | | 17.82 | 3.68 | | |
| Level of education[2] | Junior high school | 18 | 2.1 | 3.00 | 2.89 | 6.44† | p<.001 (0.03) | 6.00 | 3.58 | 4.64† | .001 (0.02) | 16.44 | 4.18 | 3.78† | .005 (0.02) |
| | High School | 157 | 22.6 | 5.29 | 3.52 | | | 6.85 | 2.87 | | | 17.22 | 3.53 | | |
| | Technical school, technical college or junior college | 181 | 25.8 | 5.06 | 3.48 | | | 6.60 | 2.89 | | | 17.52 | 3.88 | | |
| | College | 436 | 43.0 | 6.08 | 3.54 | | | 7.40 | 2.59 | | | 18.15 | 3.53 | | |
| | Graduate School | 64 | 6.2 | 6.34 | 3.59 | | | 7.72 | 2.75 | | | 18.70 | 4.03 | | |
| Occupation[1] | Health care provider | 108 | 12.6 | 7.26 | 3.03 | -5.70 | p<.001 (0.52) | 7.32 | 2.64 | -0.84 | .401 (0.09) | 18.82 | 3.70 | -2.96 | .003 (0.30) |
| | Non-health care provider | 749 | 87.4 | 5.44 | 3.58 | | | 7.08 | 2.80 | | | 17.71 | 3.66 | | |
| Biology during high school[1] | Completed | 601 | 70.1 | 6.09 | 3.42 | -3.20 | .003 (0.27) | 7.32 | 2.61 | -1.67 | .095 (0.15) | 18.02 | 3.59 | -0.65 | .518 (0.05) |
| | Uncompleted | 185 | 21.6 | 5.15 | 3.78 | | | 6.92 | 2.92 | | | 17.83 | 3.69 | | |
| Learned about genetics[1] | Yes | 219 | 25.6 | 6.77 | 3.39 | -3.92 | p<.001 (0.32) | 7.36 | 2.64 | -0.17 | .868 (0.01) | 18.62 | 3.63 | -2.81 | .005 (0.23) |
| | No | 462 | 53.9 | 5.65 | 3.50 | | | 7.32 | 2.65 | | | 17.81 | 3.42 | | |
| Presence of genetic diseases[1] | Yes | 44 | 5.1 | 7.25 | 2.49 | -3.55 | .001 (0.41) | 6.86 | 2.80 | 1.00 | .316 (0.16) | 18.66 | 3.56 | -1.15 | .250 (0.18) |
| | No | 676 | 78.9 | 5.83 | 3.55 | | | 7.28 | 2.67 | | | 18.01 | 3.62 | | |

Note: 1)unpaired t-test, 2)one-way independent ANOVA

†: F value

effect size: unpaired t-test (d), one-way independent ANOVA ($\eta^2$)

As a result of comparing CCHL scores by attribute, those with a high level of education, medical professionals, and those who had studied genetics had significantly higher CCHL scores.

## Risk perception and benefit perception for genetic testing

We confirmed the ceiling effect (mean score + standard deviation $\geqq 5$) and floor effect (mean score—standard deviation $\leqq 1$) for five items on risk perception and five items on benefit perception for genetic testing. As a result, no ceiling effect or floor effect was observed for any of the five items on risk perception or the five items on benefit perception.

Next, we performed a principal component analysis on five items for risk perception and five items on benefit perception for genetic testing. As a result, across the five risk perception items, the load of the first component was .734 to .789, and the contribution rate was 59.8%. In the five benefit perception items, the loading amount of the first component was 0.80 to 0.82, and the contribution rate was 65.1%. The total score was calculated for each of the five items on risk perception and the five items on benefit perception. Hereinafter, the total scores are referred to as the "risk perception score" and "benefit perception score." The score range is 5 to 25 points in each case.

**Table 2. The correct answer rates for genetic/genomic knowledge.**

| | | TRUE OR FALSE | correct answer | | incorrect | | don't know | |
|---|---|---|---|---|---|---|---|---|
| | | | N | % | N | % | N | % |
| 1 | One's genetic information can change over one's lifetime. | FALSE | 254 | 29.6 | 235 | 27.4 | 368 | 42.9 |
| 2 | Genetic testing may predict future disease onset. | TRUE | 569 | 66.4 | 39 | 4.6 | 249 | 29.1 |
| 3 | I share some of my genetic information with my blood relatives. | TRUE | 553 | 64.5 | 42 | 4.9 | 262 | 30.6 |
| 4 | If you have a genetic mutation (change in a gene) that is associated with a disease, you will always get the disease. | FALSE | 447 | 52.2 | 65 | 7.6 | 345 | 40.3 |
| 5 | If you have no blood relatives with a genetic disease, you do not have a genetic mutation (genetic change). | FALSE | 458 | 53.4 | 61 | 7.1 | 338 | 39.4 |
| 6 | Genetic mutations (changes in genes) that occur in somatic cells (cells other than sperm and oocytes) are passed on to the next generation. | FALSE | 161 | 18.8 | 229 | 26.7 | 467 | 54.5 |
| 7 | As genetic and genomic research advances, the probabilities and interpretations of the relationship between disease and heredity may change. | TRUE | 457 | 53.3 | 52 | 6.1 | 348 | 40.6 |
| 8 | Even if a genetic mutation (change in a gene) is found that may cause disease, there may be no effective treatment currently available. | TRUE | 473 | 55.2 | 56 | 6.5 | 328 | 38.3 |
| 9 | Knowing that a person has a genetic mutation (change in a gene) that predisposes him or her to develop a disease may lead to prevention or early detection. | TRUE | 549 | 64.1 | 44 | 5.1 | 264 | 30.8 |
| 10 | If you have a genetic mutation associated with the disease (change in a gene), it is possible that your blood relatives may have a similar mutation. | TRUE | 497 | 58.0 | 62 | 7.2 | 298 | 34.8 |
| 11 | Probability of having a child with a genetic disease is 50% means that for every four children you have, two will have the disease. | FALSE | 438 | 51.1 | 279 | 32.6 | 140 | 16.3 |

## Relationship between genomic literacy components, risk perception, benefit perception, and behavioral intention toward genetic testing

To investigate the relationships between the variables of the three components of genomic literacy, risk perception scores, benefit perception scores, and behavioral intention toward genetic testing, a correlation analysis was performed (Table 3). The partial correlation coefficient was calculated with the presence or absence of hereditary disease of the person or family as a control variable. Consequently, a significant positive correlation was found between the genetic/genomic knowledge score, the health numeracy score, the CCHL score, and the benefit perception score.

## Direct impact of components of genomic literacy on behavioral intention toward genetic testing

To investigate the effects of genomic literacy components on behavioral intention toward genetic testing, those who had a genetic disorder or a family member with a genetic disorder were excluded and a multiple regression analysis was performed, with behavioral intention

**Table 3. Correlation analysis among three components of genomic literacy in 20s and 30s.**

| Variables | Health numeracy score | CCHL score | Risk perception scores | Benefit perception scores |
|---|---|---|---|---|
| Genetic/genomic knowledge score | .524** | .220** | -.013 | .366** |
| Health numeracy score | | .211** | .076* | .277** |
| CCHL score | | | .072 | .350** |
| Risk perception scores | | | | .332** |

Note: Control variable: the presence or absence of genetic disease of the person or family

*p < .05

**p < .01

**Table 4. Multiple regression analysis with behavioral intention as the dependent variable.**

| | | Standardized partial regression coefficient | p | 95%CI |
|---|---|---|---|---|
| One's intention toward genetic testing | genetic/genomic knowledge | .109 | .005 | [.010, .057] |
| | health numeracy | .091 | .019 | [.006, .066] |
| | CCHL | .170 | p<.001 | [.031, .071] |
| | R$^2$ | .076 | p<.001 | |
| | adjusted R$^2$ | .073 | | |
| Intention to convey positive results to my partner | genetic/genomic knowledge | .086 | .021 | [.004, .044] |
| | health numeracy | .195 | p<.001 | [.044, .095] |
| | CCHL | .256 | p<.001 | [.052, .086] |
| | R$^2$ | .163 | p<.001 | |
| | adjusted R$^2$ | .160 | | |
| Intention to convey positive results to children | genetic/genomic knowledge | .046 | .226 | [-.008, .035] |
| | health numeracy | .154 | p<.001 | [.030, .085] |
| | CCHL | .238 | p<.001 | [.049, .086] |
| | R$^2$ | .112 | p<.001 | |
| | adjusted R$^2$ | .109 | | |

toward genetic testing as the dependent variable and the three components of genomic literacy as the independent variables (Table 4). Consequently, the adjusted coefficient of determination ranged from .073 to .160, which was a significant value at the 1% level for all behavioral intentions toward genetic testing. It was shown that health numeracy and CCHL among the independent variables influenced all the items on behavioral intention toward genetic testing.

## Explanation and results of the hypothetical model

We set up a hypothetical model where components of genomic literacy (genetic/genomic knowledge, health numeracy, and CCHL) influenced behavioral intentions toward genetic testing through risk and benefit perception, created a path diagram, and performed path analysis after excluding those who had a genetic disease or had a family member with a genetic disease.

The effect on "one's intention toward genetic testing" is as shown in Fig 1, but a significant path was found where genetic/genomic knowledge and CCHL affect one's intention toward genetic testing through benefit perception. However, risk perception did not directly affect one's intention toward genetic testing. Since the path coefficient between risk perception and benefit perception is high, it was interpreted that risk perception affects benefit perception and contributes to the formation of one's intention toward genetic testing. In other words, we confirmed that health numeracy affects one's intention toward genetic testing from risk perception to benefit perception.

For "one's intention toward genetic testing," the goodness-of-fit indicators showing the fit of the path diagram and data in the analysis were GFI = .994, AGFI = .960, CFI = .987, and RMSEA = .067, indicating that the model has high goodness of fit. However, the goodness-of-fit indicators of "intention to convey positive results to my partner" were GFI = .974, AGFI = .820, CFI = .925, and RMSEA = .162. The goodness-of-fit indicators of "intention to convey positive results to children" in respondents were GFI = .983, AGFI = .882, CFI = .950, and RMSEA = .128. Thus, the goodness of fit of the model was lower than that of "one's intention toward genetic testing."

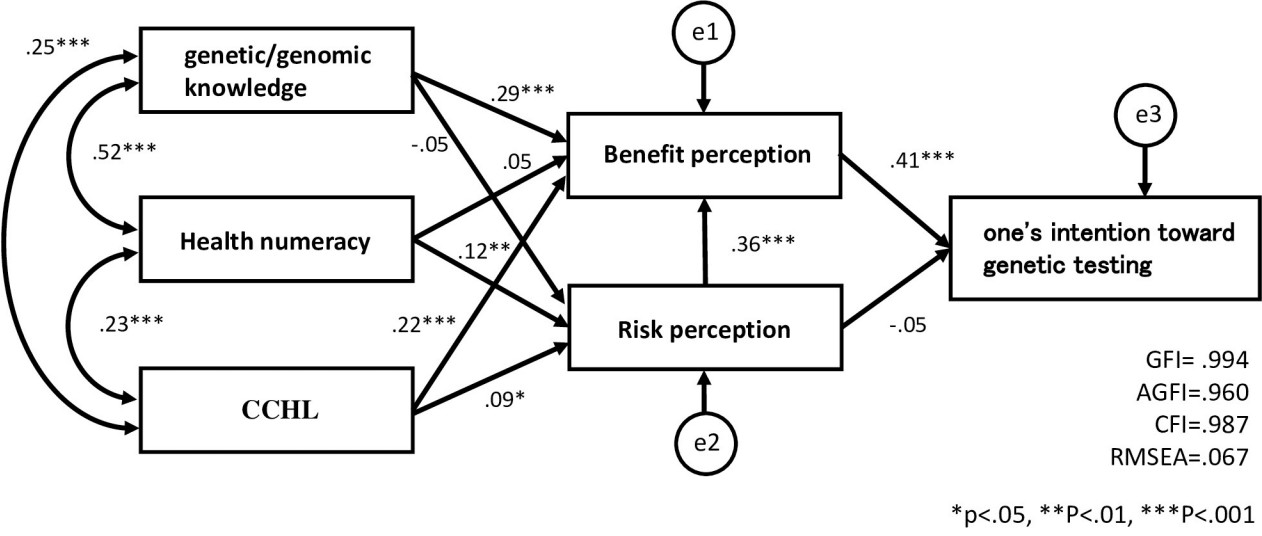

**Fig 1. Path analysis of one's intention toward genetic testing.**

## Discussion

### Relationship between genomic literacy components, risk perception, benefit perception, and behavioral intention toward genetic testing

In previous studies investigating genetic/genomic knowledge and intention toward genetic testing, those with high genetic/genomic knowledge tend to want to undergo genetic testing, and it has been shown that genetic/genomic knowledge influences decision-making [31–33]. The genetic and genomic knowledge investigated in this study also revealed that it influenced intention toward genetic testing and intention to convey positive results to partners and children. Since knowledge is considered to be one of the prerequisites for health behavior, enhancing the genetic/genomic knowledge of the general public will lead to appropriate decision-making in the genetic/genomic domain as well.

It is not only genetic/genomic knowledge, but also health numeracy and interactive and critical health literacy that influences behavioral intention toward genetic testing. Enhancing genetic/genomic knowledge leads to appropriate decision-making, and the genomic literacy scales [26, 34, 35] and evaluation methods [36] used so far have mainly only measured genetic/genomic knowledge and examined its relationship with decision-making. The results of the present study demonstrate that the three components of genomic literacy influence one's intention toward genetic testing. Therefore, it will be necessary to proceed with research on genomic literacy, including not only genetic/genomic knowledge, but also health numeracy and interactive and critical health literacy.

In addition, the genetic and genomic knowledge required for decision-making may change with rapidly advancing genomic medicine. However, health numeracy and interactive and critical health literacy are universal abilities that are not situation-dependent. From the results of this study, it can be said that health numeracy and interactive and critical health literacy are important decision-making components of genomic literacy.

### Impact of genomic literacy on behavioral intentions via risk perception and benefit perception

To clarify the process by which genomic literacy influences decision-making, a model was assumed in the present study in which the behavioral intention toward genetic testing was

influenced through risk and benefit perception based on previous studies. In other words, we created a path model of intention toward genetic testing from the components of genomic literacy through the cognitive factors of risk and benefit perception and conducted path analysis to examine it. Thus, the association of the genomic literacy component → risk and benefit perception → intention toward genetic testing was found, and the model revealed high goodness of fit. This association supported the results of previous studies regarding the decision-making process [17, 18, 37]. Therefore, it was suggested that there is a decision-making process in which the three components of each individual's genomic literacy influence their intention toward genetic testing through risk and benefit perception of genetic testing.

In this study, we reveal the path from risk perception to benefit perception to the influence on one's intention toward genetic testing. In previous studies [38–40], there was often a negative correlation between risk and benefit perception, but in the present study, there was a positive correlation between them. However, Lloyd et al. reported a positive correlation between patient risk and benefit perception for carotid endarterectomy [41], the reason being that patients recognize that high-risk surgery also has high benefits. In the present study as well, it is possible to recognize high risks and high benefits for new tests. In addition, a survey of food risk perception reported that the high numeracy group showed a positive correlation between risk and benefit perception after information provision, as compared to the low numeracy group [42]. It is suggested that those with high numeracy may consider the risks and benefits in a well-balanced manner by receiving numerical information. Therefore, the positive relationship between risk and benefit perception in the present study is considered to be a valid result.

In previous studies on numeracy, those with high numeracy made appropriate decisions based on the numbers themselves. However, people with low numeracy are susceptible to non-numerical information (emotions and moods) and cognitive bias [43, 44]. In addition, Klaczynski et al. found that numeracy facilitates rational risk decision-making in people with high critical thinking attitudes [45]. The path analysis model in the present study revealed that health numeracy did not directly affect behavioral intention toward genetic testing but did affect behavioral intention via risk perception. Increasing numeracy can lead to a better understanding of information on probability and statistics and generally better decision-making. The same is considered to be the case for behavioral intention toward genetic testing.

Many previous studies on health literacy have shown that those with high health literacy have a high intention to undergo cancer screening [46–49]. According to studies focusing on interactive and critical health literacy among the three levels of health literacy [50], people with high health literacy tend to not only have a high ability to apply knowledge to new or highly-specific health problems, but also provide information to their neighbors. As revealed in this study, CCHL influences both risk and benefit perception and has been shown to be an important component of genomic literacy. Regarding health problems related to genes and genomes, new knowledge is required due to the characteristics of genomic research, and it is necessary to update one's own knowledge. Therefore, interactive and critical health literacy can be said to be an important factor in behavioral intention toward genetic testing.

In this light, it was revealed that a model in which the three components of genomic literacy influence behavioral intention through risk and benefit perception is effective for one's own intention toward genetic testing.

We believe that our study makes a significant contribution to the literature because we clarify the three components of genomic literacy (genetic/genomic knowledge, health numeracy, interactive and critical health literacy) which, through the recognition of the benefits and risks of genetic testing affect the respondents' intention to undergo such testing.

## Limitations of this study

We created a path model of "intention to convey positive results to partners" and "intention to convey positive results to children" from the components of genomic literacy through cognitive factors of risk and benefit perception and performed path analysis. However, the goodness of fit of the model was lower than that for "one's intention toward genetic testing," indicating that the hypothetical model needs to be re-examined. The reason for this is considered to be the peculiarity of the genetic information handled in genetic testing. The peculiarities of genetic information include the fact that it is information unique to a person and does not change or cannot be changed in their lifetime (invariance), and further that there is commonality regarding such information with other individuals in a genetically-linked family [30, 51]. There is also the predictability of genetic information in that it can predict a future that was previously unknown. In other words, although an individual's genetic information is unique to that individual, it also involves the duality of already being shared by their family. Sakurai and Fukushima state that the information obtained from a single genetic test can enable early diagnosis, early treatment, or prevention of onset in relatives [52]. However, it also means that families who have no health problems may inevitably become concerned about illness or genetic problems. As an ethical issue in genetic diagnosis, Kakee points out that various forms of social discrimination due to stigma can occur if genetic testing reveals that a person is certain to have a serious hereditary disease or is likely to have a future illness [53]. For example, there is a possibility of discrimination at the time of marriage based on concerns about passing on diseases to children, information that there are patients with hereditary diseases in the family at the time of schooling or employment, or if susceptibility to the disease is known. Therefore, in addition to the components of genomic literacy, risk perception, and benefit perception, the intention to convey positive results to partners and children also needs to be considered in conjunction with other factors such as family history of the disease, the disease being genetically tested, the severity of the disease, family relationships, and values.

## Acknowledgments

We would like to thank all the participants who completed this survey. We thank Editage (www.editage.jp) for their English-language editing.

## Author Contributions

**Conceptualization:** Tomomi Miyoshi, Masaki Watanabe.

**Data curation:** Tomomi Miyoshi, Masaki Watanabe.

**Formal analysis:** Tomomi Miyoshi, Masaki Watanabe.

**Funding acquisition:** Masaki Watanabe.

**Investigation:** Tomomi Miyoshi, Masaki Watanabe.

**Methodology:** Tomomi Miyoshi, Masaki Watanabe.

**Project administration:** Tomomi Miyoshi, Masaki Watanabe.

**Resources:** Tomomi Miyoshi, Masaki Watanabe.

**Software:** Tomomi Miyoshi.

**Supervision:** Tomomi Miyoshi, Masaki Watanabe.

**Validation:** Tomomi Miyoshi, Masaki Watanabe.

**Visualization:** Tomomi Miyoshi, Masaki Watanabe.

**Writing – original draft:** Tomomi Miyoshi.

**Writing – review & editing:** Tomomi Miyoshi, Masaki Watanabe.

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
