## [Decision Letter · Decision Letter 0]

8 Feb 2023

PONE-D-22-23128Impact of genomic literacy components on genetic testing decision-making in the general Japanese population in the 20s and 30sPLOS ONE

Dear Dr. Watanabe,

Thank you for submitting your manuscript to PLOS ONE. I apologize for the delay in the review process.After careful consideration, we feel that it has merit but does not fully meet PLOS ONE’s publication criteria as it currently stands and requires a few minor revisions. Therefore, we invite you to submit a revised version of the manuscript that addresses the points raised during the review process.

 Please ensure that the data associated with the manuscript or its supporting information are available via a public repository following PLoS One guidelines and update the methodology section to address the concerns raised by Reviewer #1. In addition, please update Figure 1 and Table 1 and check that all references are complete and correct. 

We look forward to receiving your revised manuscript.

Kind regards,

Qasim Ayub, Ph.D.

Academic Editor

PLOS ONE

Journal Requirements:

Reviewers' comments:

Reviewer's Responses to Questions

**Comments to the Author**

1. Is the manuscript technically sound, and do the data support the conclusions?

Reviewer #1: Yes

Reviewer #2: Yes

2. Has the statistical analysis been performed appropriately and rigorously? 

Reviewer #1: Yes

Reviewer #2: Yes

3. Have the authors made all data underlying the findings in their manuscript fully available?

Reviewer #1: Yes

Reviewer #2: No

4. Is the manuscript presented in an intelligible fashion and written in standard English?

Reviewer #1: No

Reviewer #2: No

5. Review Comments to the Author

Reviewer #1: Please check the attachment for details:

Page 4: Better to cite some related studies in Japan and then bring up the research gap.

Page 5: The authors need to mention how they calculated the sample size and the sampling technique used to draw this sample.

Page 8: There is no information about testing correlation and regression analysis assumptions. Authors must provide this information.

Page 11: What is the p-value here?

Page 14: It would be better if the authors provided a few subtitles for this section and then discussed the individual objectives under each subtitle.

Reviewer #2: Fig 1 is referred to in the text. Unfortunately, the Fig was presented in the manuscript. This made it difficult to follow and evaluate all aspects of the manuscript that made reference to it.

Overall the manuscript is in good shape. The methodology is robust and the statistical analysis is sound. However, there are a few minor errors in the manuscript. Some of these are shown below:

Line 78-19 Please provide references pertaining to the many authors.

Line 117 to read "Questionnaire items".

Line 213 Table 2 does not show any % contrary to the claim made in line 213. The authors should attend to the formatting of the table so that it reflects all the data. Although the percentages are mentioned in the text, it is important to make sure that the Table is complete.

6. PLOS authors have the option to publish the peer review history of their article (what does this mean?). If published, this will include your full peer review and any attached files.

Reviewer #1: No

Reviewer #2: No

---

## [Author Response · Author response to Decision Letter 0]

2 Mar 2023

Dear Dr. Ayub,

Thank you very much for your e-mail and review of the manuscript (PONE-D-22-23128) that we sent on 20th August 2022. We sincerely thank the two reviewers for providing constructive comments regarding the improvement of the original manuscript. Thus, it is with great pleasure that we resubmit our article for further consideration. We have incorporated changes that reflect the detailed suggestions you have graciously provided. We also hope that our edits and the responses we provide below satisfactorily address all the issues and concerns you and the reviewers have noted. To facilitate your review of our revisions, the following is a point-by-point response to the questions and comments delivered in your letter dated 23rd December 2022.

---

## [Editor Report · Decision Letter 1]

10 Mar 2023

Impact of genomic literacy components on genetic testing decision-making in the general Japanese population in the 20s and 30s

PONE-D-22-23128R1

Dear Dr. Watanabe,

We’re pleased to inform you that your manuscript has been judged scientifically suitable for publication and will be formally accepted for publication once it meets all outstanding technical requirements.

Kind regards,

Qasim Ayub, Ph.D.

Academic Editor

PLOS ONE

---

## [Editor Report · Acceptance letter]

14 Mar 2023

PONE-D-22-23128R1 

Impact of genomic literacy components on genetic testing decision-making in the general Japanese population in the 20s and 30s 

Dear Dr. Watanabe:

I'm pleased to inform you that your manuscript has been deemed suitable for publication in PLOS ONE. Congratulations! Your manuscript is now with our production department. 

Kind regards, 

on behalf of

Dr. Qasim Ayub 

Academic Editor

PLOS ONE